# Physical Activity, Body Mass, and Adherence to the Mediterranean Diet in Preschool Children: A Cross-Sectional Analysis in the Split-Dalmatia County (Croatia)

**DOI:** 10.3390/ijerph16183237

**Published:** 2019-09-04

**Authors:** Lejla Obradovic Salcin, Zeljka Karin, Vesna Miljanovic Damjanovic, Marko Ostojic, Andrea Vrdoljak, Barbara Gilic, Damir Sekulic, Maja Lang-Morovic, Josko Markic, Dorica Sajber

**Affiliations:** 1Clinic for Physical Medicine and Rehabilitation, University Hospital Mostar, Bosnia and Herzegovina, 88000 Mostar, Bosnia and Herzegovina; 2Faculty of Health Sciences, University of Mostar, 88000 Mostar, Bosnia and Herzegovina; 3Teaching Institute of Public Health of Split Dalmatian County, 21000 Split, Croatia; 4Faculty of Kinesiology, University of Split, 21000 Split, Croatia; 5Croatian Institute of Public Health, Health Promotion Division, 10000 Zagreb, Croatia; 6Department of Pediatrics, University Hospital of Split, 21000 Split, Croatia; 7School of Medicine, University of Split, 21000 Split, Croatia; 8Faculty of Sport, University of Ljubljana, 1000 Ljubljana, Slovenia

**Keywords:** diet quality, pediatrics, public health, body composition

## Abstract

Physical activity, body mass, and dietary habits are known to be important determinants of overall health status, but there is an evident lack of studies that examine these issues specifically in preschool children. The aim of this study was to identify associations that may exist between adhering to the Mediterranean diet (MD), levels of physical activity (PA), and body composition indices in apparently healthy preschool children from southern Croatia. Participants were 5- to 6-year-old preschoolers from the Mediterranean part of the country (the Split-Dalmatia County; n = 260, 126 females). Adherence to the MD was observed by the Mediterranean Diet Quality Index (KIDMED), PA level was evaluated by the Preschool-age Children’s Physical Activity Questionnaire (Pre-PAQ), and responses were collected from the parents. The participants’ waist circumferences (in cm), waist-to-hip ratios, and body mass index (in kg/m^2^, and in a z-score calculated relative to the normative value for age and sex) were used as indicators of body composition. All children were of the same age and tested over a one-month period of the same year as a part of the regular examination undertaken before attending elementary school. With only 6% of the children having a low KIDMED score, adherence to the MD was high. MD adherence was higher in girls (Chi-square = 15.31, *p* < 0.01) and children who live on the coast of the Adriatic Sea (Chi-square = 18.51, *p* < 0.01). A mixed effects logistic regression (with kindergarten as random factor) identified sedentary activity to be negatively associated with MD adherence (OR per point: 0.65, 95% CI: 0.44–0.91). High adherence to the MD in the studied sample may be attributed to regulated feeding in kindergarten. Considering that most Croatian elementary schools do not provide food to their students, MD adherence should be investigated later in life and also in other parts of the country where the MD is culturally less prevalent.

## 1. Introduction

The human body is designed such that most of its systems (cardiovascular, metabolic, skeletal, muscle, etc.) cannot develop and function optimally if they are not stimulated by regular physical activity [1]. Moreover, it is well established that physical activity (PA) provides numerous health benefits, such as assisting in the prevention of chronic diseases, balancing energy expenditure, and maintaining a healthy body composition [2]. Additionally, PA increases functional capacities and can reduce cardiovascular risks [3]. Finally, regular and appropriate PA is an important component for normal growth and the development of the aerobic capacity, muscular strength, flexibility, and motor skills of children and adolescents [4,5].

The World Health Organization (WHO) recommends a minimum of 60 min of “moderate- to vigorous-intensity PA per day” for preschool children, at least 3 times per week [6]. However, it is evident that children are currently insufficiently active and do not achieve PA recommendations [7]. As a result, an increasing number of children are obese and overweight. Further, a lack of physical activity negatively affects overall health [8]. Studies performed with children have confirmed that screen time (i.e., time spent watching TV and playing video games) and inactivity levels are strongly associated with obesity risk and increased levels of body fat even in very young children [9,10]. Finally, the patterns of PA from childhood continue to adulthood, so young children should be encouraged to be involved in some kind of PA or play not only to prevent becoming obese or overweight but also to maintain good health later in life [11].

Apart from PA, type of diet is another important factor contributing to one’s overall health status. While the Western diet (i.e., a diet rich in saturated fats and red meats and low in fresh fruits and vegetables) has been linked to numerous negative health-related consequences (i.e., hypertension, hypercholesterolemia, heart disease, and obesity), one of the healthy diets is the so-called Mediterranean diet (MD) [12,13,14]. In brief, an MD eating plan is based on typical foods and recipes found in Mediterranean-style cooking, including the use of olive oil, the consumption of vegetables, fruits, whole grains, nuts and seeds, and the moderate consumption of legumes and red wine [15]. Priority for animal products is given to fish and white meat over processed and red meat. This dietary pattern is related to a balanced and varied diet, providing most of the necessary macronutrients in the right proportion. Compared to the Western diet, the MD contains fewer saturated fatty acids and more monounsaturated fatty acids, as well as a high content of complex carbohydrates and fibers and an important amount of antioxidants [16]. Studies have confirmed that adherence to the MD is marked by numerous health benefits, such as the prevention of cardiovascular diseases, diabetes mellitus, hypertension, inflammation, cancer, depression, asthma among children, and cognitive decline in adults [17,18].

While both PA and type of diet are important determinants of health status, studies have already examined the possible associations between MD adherence and PA in children. In brief, a Spanish study that investigated 11- to 12-year-old children reported an association between MD adherence and lifestyle habits, with MD adherence being positively associated with PA and negatively associated with screen time (r = 0.38 and −0.18, respectively) [19]. Another Spanish study, which observed adolescents (11–18 years of age) from the southern part of the country, found greater MD adherence in more physically active adolescents [20]. Similar conclusions were provided in the Greek study conducted with 10- to 12-year-old children, where authors reported higher MD adherence in children who had higher PA levels [21]. Moreover, a Chilean study demonstrated a positive association between physical fitness and MD adherence in 10-year-old children [22]. Collectively, it seems that PA is positively associated with MD adherence, but there is an evident lack of studies that investigate the association between PA and MD adherence, specifically in preschool children [23].

Croatia is a country on the Balkan Peninsula and is one of the former Yugoslav Republics. Because the country is located along the Adriatic Sea, a substantial part of Croatia is oriented toward Mediterranean culture and traditions and, consequently, to the MD. However, the results of recent studies did not confirm a high adherence to the MD in Croatia. Specifically, a study evaluating dietary patterns among adults (>18 years old) from southern Croatia revealed poor compliance with the recommendations for the consumption of the MD [18]. Several potential reasons were proposed for this low adherence to the MD. The first reason is finances, as Mediterranean products cost more than Western products. Additionally, fast food is becoming highly available, while Mediterranean cuisine requires preparation and is, therefore, relatively time-consuming. On the other hand, a study investigating the Mediterranean Diet Quality Index (KIDMED) scores in Croatian university students found that 10.5% of students had low scores, 46.7% had average scores, and 42.8% had good results [24].

Physical activity and type of diet are known to have a significant influence on health in general, but there is a lack of studies examining the associations between physical activity and type of diet in Croatia and southeastern Europe. Furthermore, to the best of our knowledge, no study in the region has examined these issues specifically in preschoolers, who are generally the most vulnerable group since children develop their life-long habits and behaviors in this period of life. Therefore, the aim of this research was to determine the PA levels and adherence to the MD and to investigate the associations among PA, MD adherence and indices of body build and body composition in healthy preschool children. Initially, we hypothesized that adherence to the MD is positively associated with physical activity in preschool children.

## 2. Materials and Methods

### 2.1. Participants

The participants in this study were preschoolers (n = 260; 5–6 years of age; 126 girls) from three municipalities in the Split-Dalmatia County in southeastern Croatia. All of the children were tested during their mandatory medical examination, which was held during March and April 2017, 5–7 months before school enrollment (September 2017) and the beginning of the primary school education. It is important to note that the sample actually comprised randomly selected children because one year of kindergarten education prior to elementary school (preparatory school) is mandatory for all children in Croatia. Additionally, all children observed in this study were of the same age (in the last year of kindergarten), and all children were tested within a time frame of two months. Therefore, the possibility that the sample of participants involved several children from one family is negligible. Since the study involved three municipalities in the studied County (Split, Solin, and Imotski), two of which are located strictly on the Mediterranean Coast (i.e., Split and Solin), and one of which is geographically located in the continental part of the county (Imotski), the children were grouped accordingly (Coast vs. Inland). The protocols were approved by the Ethical Board of the School of Medicine of the University of Split, Split, Croatia (EBO: 2181-198-03-04-16-0009). One of the investigators informed the parents of the study purposes and potential benefits and risks of participation. For the purpose of this study, parents responded to questionnaires (see later text for details) for their children. The response rate was high (84%), but we included only results of children with no evident health-related problems (i.e., children who were identified as healthy with regard to their inclusion in elementary school education 5–6 months after this study). Most specifically, of the children observed in this study, 2 boys were excluded from the study (due to psychological disorders that prevented them from being involved in regular scholastic programs). Almost all of the parents who were observed as “nonresponders” (16%, see previous text) initially agreed to participate in the study but were not able to objectively respond to questionnaires on physical activity and/or MD adherence because they were not fully aware of their child’s habits (i.e., they had no knowledge about the dietary habits of their child in kindergarten, and/or they were not familiar with child’s PA outside of the home environment).

Based on (i) an estimated 20% incidence of a poor KIDMED score [24], (ii) a population sample of approximately 2500 preschoolers in the studied region (Split-Dalmatia County), (iii) a margin of error of 0.05, and (iii) a confidence level of 0.95, the required sample size for this investigation was calculated to be 224 participants.

### 2.2. Variables

The variables in this study included sex, location (Coast vs. Inland), anthropometric indices (body mass index (BMI), waist circumference (WC), and waist-to-hip ratio (WHR)), PA level (obtained by the Preschool-age Children’s Physical Activity Questionnaire (Pre-PAQ)), and adherence to the MD (obtained by KIDMED).

Anthropometric indices: BMI was calculated from the measured body height (BH) and body mass (BM) using the following equation: BMI = BM (kg)/BH^2^ (m) [25]. Additionally, to demonstrate the actual participants’ status in BMI levels, we calculated the BMI z-score (a measure of how many standard deviations above or below the Croatia normative data for a specific age and sex) [26,27]. The WC (cm) was measured at the midway point between the rib cage margin and iliac crest when the subjects were standing, using an anthropometric tape [28]. The WHR (cm) was calculated as the WC divided by the hip circumference. The hip circumference (cm) was measured with the anthropometric tape as the distance around the greater trochanters at the widest point [29]. All anthropometric measures were performed by experienced technicians by using standardized measuring equipment (Seca Instruments Ltd., Hamburg, Germany). The measurements of WC and hip circumference were performed over three consecutive trials, and the median result was used as the final score for each participant.

Physical activity level: The Pre-PAQ is a questionnaire that intends to measure habitual physical activity and time spent in sedentary behaviors in the home environment of a preschool child over three days (one weekday and two weekend days). The Pre-PAQ categorizes activities into five levels: L1 is stationary with no movement, L2 is stationary with limb or trunk movement, L3 is slow activity, L4 is medium activity, and L5 is fast-paced activity [30]. In this study, L1 and L2 are combined to represent stationary activity. The Pre-PAQ was professionally translated into the Croatian language, and for this investigation, it was completed by the child’s parent [31].

Level of adherence to the MD: The KIDMED test evaluates adherence to the MD among children and adolescents aged from 2 to 24 years. The KIDMED consists of 16 questions: Questions denoting a positive connotation with respect to the MD are marked as +1, while questions with negative aspects are marked as −1. The index theoretically ranges from −3 to 12, and scores are classified into three levels: (i) low MD adherence: KIDMED score ≤ 3; (ii) average MD adherence: score 4–7; and (iii) good MD adherence: score ≥ 8 [16]. Questions in the KIDMED are as follows: eats one serving of fruit or drinks fruit juice every day (+1); eats a second serving of fruit every day (+1); eats fresh vegetables (salads) or cooked vegetables once a day (+1); eats fresh vegetables or cooked vegetables more than once a day (+1); eats fish regularly (at least 2 or 3 times/week) (+1); visits a fast-food establishment at least once a week (−1); likes vegetables (+1); eats pasta or rice almost daily (>5 days/week) (+1); eats cereals or flour derivatives (bread, etc.) for breakfast (+1); eats dried fruit regularly (at least 2 or 3 times/week) (+1); uses olive oil at home (+1); does not eat breakfast (−1); eats a dairy product (milk, yogurt, etc.) for breakfast (+1); eats mass-produced cakes for breakfast (−1); eats two portions of yogurt and/or 40 g of cheese daily (+1); and eats sweets and confectioneries several times a day (−1) [32]. The questionnaire was previously adopted for the Croatian language and tested for reliability and validity [24]. In short, test–retest Kappa statistics showed moderate to excellent agreement for responses to single questions (ranging from 0.44 to 0.93), and moderate correlations between KIDMED results obtained on two testing occasions (ranging from 0.59–0.61) [24]. Although participants were originally grouped according to their KIDMED score into three groups (low-, average-, and high-MD adherence), for the purpose of some statistical analyses (please see the following text), the low- and average-MD adherence groups were grouped together (L/A-MD) and compared with the high-MD adherence (H-MD) group.

### 2.3. Statistics

All data were log-transformed to reduce the non-normality of errors [33]. The descriptive statistics included means and standard deviations (for normally distributed variables), medians and 25th to 75th percentiles (for PA-levels), and percentages (for categorical variables). Although log-transformed values were used in the statistical analyses, in the following results, we reported the real (e.g., nontransformed) values for all variables.

The MD adherence status for boys and girls was compared by the Chi-square test. The linear mixed model (with kindergartens as the random factor) was calculated to identify the main effects (sex, MD adherence) and interactions (sex x MD adherence) in the studied variables. The distribution of residuals was checked by the Kolmogorov Smirnov test and were found to be normally distributed for all dependent variables except for BMI.

The mixed model logistic regression (with kindergarten as random factor) was applied to demonstrate the correlates of MD adherence (observed as binomial criterion: L/A-MD [coded as “1”] vs. H-MD [coded as “2”], as suggested previously [34]). The regression calculations included the nonadjusted model (Model 1), model adjusted for sex (Model 2), and model adjusted for sex and location (coast-inland). The goodness of fit was evaluated by the Hosmer–Lemeshow test (a statistically significant result indicates that the model does not adequately fit the data).

The statistical significance of *p* < 0.05 was applied, and Statistica ver. 13 was used for all calculations (TIBCO Software Inc., Palo Alto, CA, USA).

## 3. Results

Approximately 6% of the children had a low KIDMED score (low MD adherence), 24% had average MD adherence, and almost 70% of the children were classified as having high MD adherence. Adherence to the MD was higher for girls than for boys (Chi-square: 15.31; *p* < 0.01). Additionally, children who live in coastal municipalities had higher adherence to the MD than their peers who were residentially located inland (Chi square: 18.51, *p* < 0.01) (Figure 1).

The most evident differences between girls and boys in responses to the KIDMED questionnaire were found for eating fresh/cooked vegetables once a day (65% and 50%, respectively; Chi-square: 6.03, *p* = 0.01), eating a second serving of fruit every day (48% and 35%, respectively; Chi-square: 4.21, *p* = 0.04), consumption of legumes (Chi-square: 4.72, *p* = 0.03; 15%, and 7% of the responses were affirmative for girls and boys, respectively) (Figure 2).

The main effects of MD adherence were significant for L1 (*F*-test: 22.23, *p* = 0.03) with lower sedentary activity in children with good MD adherence (Table 1).

The mixed effect logistic regression Model 1 (with kindergarten as random factor), with MD adherence as the dependent variable, showed a significant association between WC and the criterion (OR per cm: 0.90, 95% CI. 0.81–0.97, *p* < 0.01) and L1 (OR per point: 0.65 95% CI: 0.48–0.98, *p* = 0.03). Specifically, H-MD adherence was found in children with a low WC and low sedentary activity. When adjusted for sex (Model 2), a significant predictor of MD adherence was L1 (OR per point: 0.64, 95% CI: 0.45–0.91, *p* = 0.02) (Table 2).

Regression Model 3 (adjusted for sex and location) identified L1 (OR per point: 0.65; 95% CI: 0.44–0.91, *p* = 0.02) as a factor significantly associated with MD adherence, with a higher likelihood for H-MD adherence in children who have low L1 scores. The remaining variables were not significantly associated with MD (OR [95% CI]; BMI-Z per z-score: 1.13 [0.74–1.76], WHR per score: 0.47 [0.09–2.30], WC: 0.94 [0.83–1.05], L2 per point: 1.46 [0.90–2.34], L3 per point: 1.76 [0.97–2.92], L4 per point: 0.90 [0.56–1.48], and L5 per point: 0.84 [0.36–1.86]) (Figure 3). The Hosmer–Lemeshow test results (Chi square = 6.31, 7.04, 9.52, *p* = 0.62, 0.55, 0.30 for Model 1, Model 2, and Model 3, respectively) indicate that the goodness of fit was satisfactory.

## 4. Discussion

There are some important findings in this study, which will be discussed subsequently. First, the results showed a relatively high prevalence of MD adherence in the studied preschool children. Second, children who had high KIDMED scores were less likely to be sedentary. Collectively, we may confirm our initial study hypothesis.

### 4.1. Adherence to the Mediterranean Diet in Croatian Preschoolers

Adherence to the MD is rarely studied in Croatia, and, to the best of our knowledge, there is no study that has specifically studied this issue among preschool children. However, studies of other populations showed variable results. In brief, while a study of Croatian adults showed poor adherence to the MD, the results from a study of university students showed somewhat better results, with almost 90% of participants reporting average and high adherence to the MD [18,24]. Our results show encouraging figures, with only 6% of the preschoolers having a low KIDMED score.

For the purpose of comparing the results, studies performed with children in other Mediterranean countries are particularly interesting [21,32,35,36]. In a study of Spanish children and adolescents, a low KIDMED score was found for only 4.2% of the studied participants, while 49.4% of the participants had medium adherence, and 46.4% of the participants had high results [32]. In contrast, two studies observing Greek schoolchildren aged 10–12 years found that less than 10% of children had an optimal KIDMED score [21,37]. Supporting later results, research in Cyprus revealed that only 6.7% of children aged 9–13 years were classified as having high adherence to the MD, whereas 37% of children had low MD adherence [35]. Additionally, an investigation that included Italian primary school children (8–9 years old) showed that only 5% of the sample had an optimal KIDMED score, while 62% and 32% of the sample had medium and low scores, respectively [36]. In most cases, low adherence to the MD is explained by a low consumption of fruits and vegetables.

In our study, a low KIDMED score was found in approximately 6% of the children, which corroborates the results of the previously cited studies that reported high adherence to the MD in Spanish children [19,32]. As evident by specific data about affirmative responses on the KIDMED questionnaire, it seems that such results are mostly influenced by a high consumption of dairy products (89%), a high usage of olive oil at home (77%), consumption of a breakfast that includes cereals (81%), and regular consumption of fruits (88%) (even two servings of fruits per day (48%)). Additionally, a low consumption of sweets, candies, and fast food likely contributed to the relatively high adherence to the MD in preschoolers observed in this study (Figure 2).

Although we have no exact evidence, we assume that such encouraging figures can be explained by the participants’ ages and by the fact that nearly all children observed in this study are attending kindergarten and, therefore, follow institutionalized nutritional recommendations. Specifically, Croatian kindergartens follow specified nutritional programs that promote an increased amount of fruits, vegetables, fish, meat, and dairy products by avoiding foods high in saturated fats and sugar [38]. The children we observed were in the last year of preschool education. Therefore, they spent most of their days in a kindergarten and were exposed to healthy dietary habits. At this age, this exposure probably translates into nutritional habits at home [39]. This explanation is plausible even if we discuss the contradictory findings of previous studies performed in Croatia, where the authors reported a low prevalence of MD adherence in adults and a much higher adherence in college students [18,24]. In short, college students regularly eat at student restaurants and, in most cases, consume predefined menus, which frequently include MD foods.

Regardless of the influence of institutionalized feeding, parents are probably one of the most influencing factors on MD adherence. In short, young children (i.e., preschoolers) are under the strong influence and supervision of their parents, which almost certainly translates to their (healthy) eating habits. Such a mechanism (i.e., social influence) was recently proposed as one of the explanations for the higher MD adherence in primary-school students than in secondary-school students from Spain [34]. In short, older children are more independent of their parents and, therefore, are less likely to follow appropriate nutritional plans. Additionally, the availability of pocket money almost certainly increases the likelihood of fast-food consumption, which is an additional factor that negatively influences nutrition. In contrast, in preschoolers, most of the factors that could negatively influence nutrition are minimized, which altogether results in the previously discussed encouraging figures and MD adherence in this age group. However, the question of what happens with children’s eating habits in the following years, especially considering that most of the Croatian primary schools do not provide institutionalized feeding for their students, remains unanswered.

### 4.2. Mediterranean Diet and Physical Activity

Our results showed that children with high KIDMED scores were less likely to be sedentary. Likewise, previous studies reported a positive association between PA and KIDMED scores in Greek and Spanish children (ages 4–12 and ages 11–12, respectively) [19,21]. Furthermore, the Spanish study reported that children who displayed greater adherence to the MD had less screen time and other sedentary behaviors, which directly supports the negative association between L1 and KIDMED scores in our investigation [19]. These results collectively indicate that children not only follow a healthy diet but also have a healthier lifestyle in general. However, the previously cited studies included either a relatively large age span (ages 4–12) or older children (ages 11–12) [19,21], and we examined a very narrow age range in a group of 5- to 6-year-old preschoolers. In view of the lack of studies evaluating such samples of participants, our results of the positive association between MD adherence and PA in preschoolers are novel to some extent.

Although the cross-sectional study design does not allow for an explanation of the cause–effect relationship between PA and KIDMED scores, we can speculate about three possible mechanisms underlying the relationships between these variables. First, it is possible that MD adherence influences the PA of preschoolers by allowing those children who eat properly to be more physically active simply by providing them with the necessary nutrients. Indeed, nutrition quality is vital for providing energy and achieving good health and physical performance. It is important to acquire enough nutrients to allow children to meet their needs for growth and tissue maintenance, which will altogether allow them to meet the metabolic demands of PA, especially with regard to vigorous PA [40].

Second, it is also possible that PA indirectly influences MD adherence through specific behavioral and environmental mechanisms. Briefly, those children who are more active are more likely to be outside of the home (for playground activities, sports, etc.), and, therefore, they are not in a position to eat snacks and lean more toward proper dietary habits (regular meals consisting of MD food). While preschoolers are not financially independent, it is unlikely that they will consume fast food when alone in out-of-home situations (i.e., on the street). This phenomenon is particularly possible because we studied preschoolers who are “institutionalized” most of the time and eat mostly in kindergartens (for details see previous text).

Third, and most likely, it is possible that those children who are physically active and avoid sedentary habits are actually properly educated, mentored and instructed by their parents. These children are therefore simultaneously oriented toward various positive life habits, which include both proper eating and lower levels of sedentary activity [13]. These habits are most likely associated with their socioeconomic status (SES). Indeed, previous studies found SES to be one of the most important determinants of both MD adherence and PA in children [19,23]. Regrettably, in this study, we did not collect the data about SES, but this choice was made intentionally. In short, testing was not anonymous, and parents responded for their children, so we tried to avoid questions that could possibly have negative connotations among parents. Additionally, a possible clustering of the data according to differences among kindergartens (i.e., state-financed vs. private) was also limited because more than 95% of all children in Croatia are involved in standard preschool education programs that are directly financed by County-governments (i.e., state kindergartens) and are affordable. Additionally, private kindergartens are rarely the first choice of Croatian families since parents regularly apply for a state kindergarten first, and if the application is not successful (mostly because of the limited number of places available for the afternoon shift), they decide to include their child in a more flexible and more expensive private kindergarten [41].

Although the associations between PA and MD adherence identified here are generally supportive of previous investigations, certain differences in the methodological approaches between our study and previous studies should be highlighted. Namely, previous studies investigated older children and used different PA questionnaires (e.g., PAQ-C), which score PA on one scale (e.g., a higher numerical value simply presents the higher PA) [19]. Additionally, our study is one of the first studies in younger preschool children assessing PA levels by the Pre-PAQ [30]. Specifically, the advantage of the Pre-PAQ is that children’s activity levels are divided into different categories and scored independently for different types of PA, from sedentary to vigorous PA. In contrast, other questionnaires do not discriminate between specific activity intensities and provide a summary activity score, which fails to specify how long the children are active and how long they are engaged in sedentary behaviors [42]. Since our study found specific associations between different PA levels and KIDMED scores for L1, which was negatively associated with KIDMED scores, and L3, which was positively associated with KIDMED scores, this result highlights the specific applicability of the Pre-PAQ in studying this problem and similar problems in children and adolescents.

### 4.3. Limitations and Strengths

The main limitation of this study is that some variables observed (e.g., PA and MD adherence) were not objectively measured but were reported by the parents of the children. Therefore, parents may lean toward socially desirable answers (i.e., higher levels of PA and higher prevalence of MD adherence). Second, generalization of the results is limited since we observed only one part of the country (e.g., the Split-Dalmatia County in southern Croatia) where people lean more toward the Mediterranean lifestyle not only because of the typical Mediterranean climate but also because of cultural and traditional reasons (i.e., consumption of fish and olive oil is common on the coast of the Adriatic Sea). Therefore, it is questionable whether the results are applicable to other continental parts of the country. Finally, in this study, we were not able to observe some possibly important confounding variables that could directly or indirectly influence the observed associations (i.e., the socioeconomic status of the family). The non-anonymity of the testing did not allow us to collect data about socioeconomic status, and in future studies, this problem may be overcome by observing other characteristics known to be associated with the socioeconomic status of a family (e.g., parental education, parental employment, and characteristics of the living environment). Finally, we must mention that this study included many statistical tests, and therefore, there is a possibility that some statistically significant findings are potentially a result of “type I error”. As a result, the findings from this study should be replicated in future investigations.

The main strengths of the study are related to the specific sample of participants. Indeed, to the best of our knowledge, this is the first study in Croatia and one of the first studies in the region in which a sample of participants consisted exclusively of preschool children and focused on a very narrow age range (e.g., children attending the last year of preschool education). Additionally, all measurements were performed in the standardized facilities, and evaluations were performed in the same period of the year (March–April 2017). Therefore, we believe that our results, although not fully comprehensive, will improve the level of knowledge about these important public health problems.

## 5. Conclusions

This study revealed high adherence to the MD in Croatian preschool children. The authors of the study are of the opinion that this result, although encouraging, is mostly influenced by the institutionalized eating of the children involved in the study (i.e., most children attend kindergarten and consume planned meals). On the other hand, this situation will probably change later in life since organized school feeding in Croatia is very rare, and the vast majority of primary schools in the country do not provide regular meals for their students. Therefore, it is questionable whether older children (i.e., those attending primary schools) are also engaged in healthy lifestyles. As a result, it would be important to determine the dietary habits of school-age children.

We have found a negative association between sedentary PA and MD adherence. However, the cross-sectional design of the study does not allow us to clearly describe the causality between the variables. Therefore, further studies are needed to establish the true cause–effect relationships between the studied variables.

## Figures and Tables

**Figure 1 ijerph-16-03237-f001:**
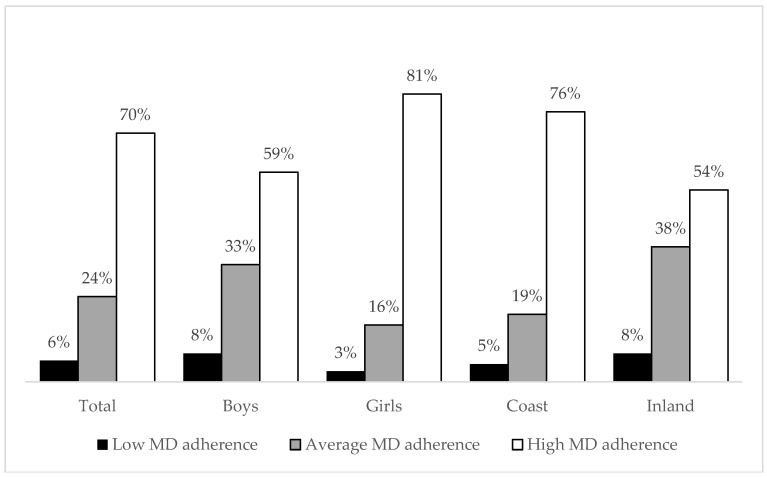
Adherence to the Mediterranean diet (MD) in preschoolers from southern Croatia.

**Figure 2 ijerph-16-03237-f002:**
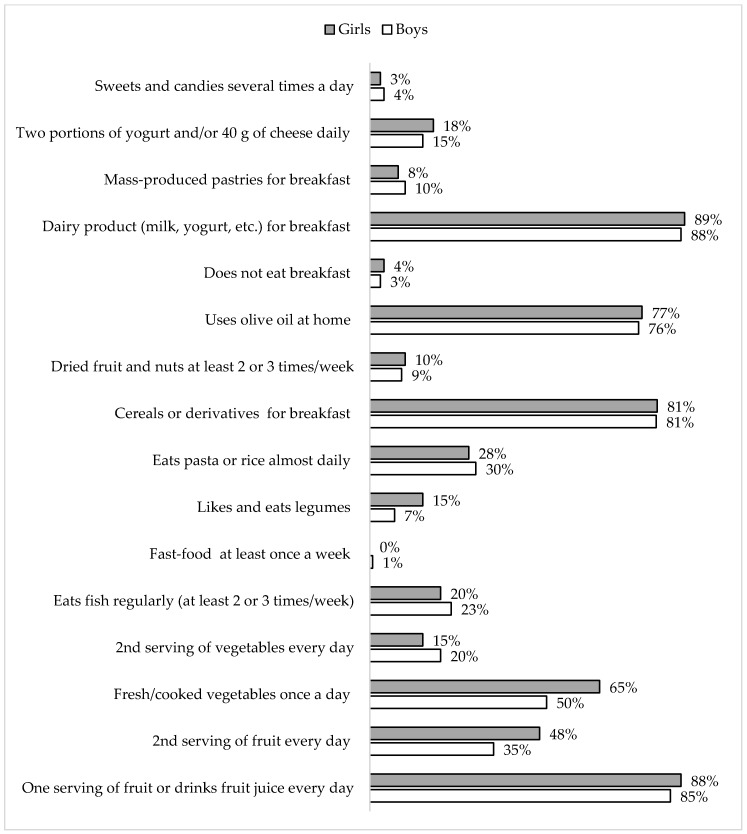
The percentage of affirmative answers to each of the 16 Mediterranean Diet Quality Index (KIDMED) questions in boys and girls.

**Figure 3 ijerph-16-03237-f003:**
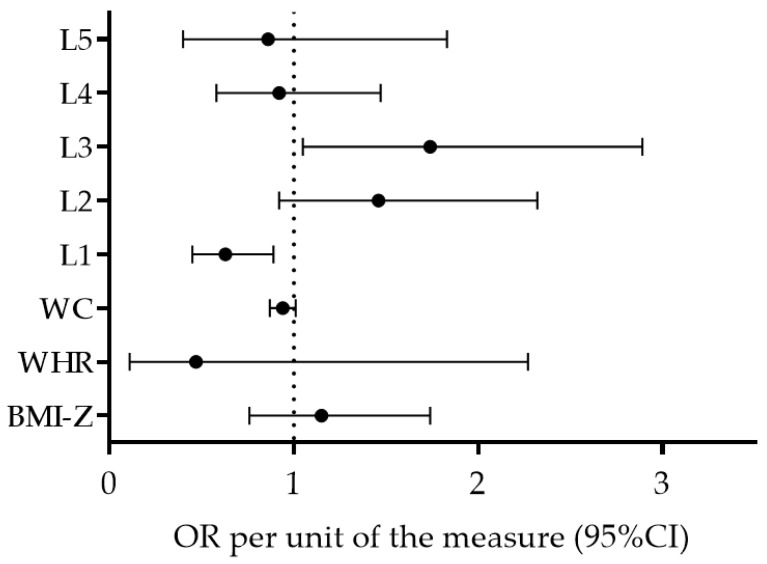
Factors associated with MD adherence in preschoolers—logistic regression model adjusted for geographical location (Coast–Inland) and sex.

**Table 1 ijerph-16-03237-t001:** Anthropometric indices and physical activity level in preschoolers with results of the linear mixed model.

	Sex		MD		ANOVA		
	Boys	Girls	L/A-MD	H-MD	Sex	MD	Sex x MD
					F (*p*)	F (*p*)	F (*p*)
BMI (kg/m^2^) *	15.90 ± 2.14	15.91 ± 2.19	15.91 ± 2.63	15.89 ± 1.94	0.01 (0.99)	0.05 (0.84)	0.93 (0.51)
BMI-Z (z-score) *	−0.04 ± 0.99	0.14 ± 0.98	0.04 ± 1.20	0.05 ± 0.88	3.77 (0.79)	0.05 (0.84)	0.98 (0.51)
WHR (ratio) *	0.88 ± 0.05	0.87 ± 0.05	0.88 ± 0.05	0.87 ± 0.05	3.57 (0.22)	0.08 (0.79)	0.02 (0.89)
WC (cm) *	57.25 ± 7.39	57.15 ± 5.69	58.47 ± 6.09	56.67 ± 6.74	3.01 (0.30)	3.93 (0.22)	1.37 (0.46)
L1 (score) ^¥^	1.0 (0.5–1.5)	1.2 (0.5–1.67)	1.1 (0.67–1.5)	1.0 (0.5–1.5)	0.07 (0.81)	22.23 (0.03)	1.01 (0.41)
L2 (score) ^¥^	0.67 (0.33–1.17)	0.5 (0.25–1.08)	0.67 (0.33–1.17)	0.67 (0.33–1.17)	9.01 (0.16)	1.41 (0.42)	0.03 (0.87)
L1+L2 (score) ^¥^	1.83 (1.17–2.67)	1.67 (1.17–2.5)	1.83 (1.33–2.67)	1.67 (1.17–2.5)	0.12 (0.76)	3.05 (0.34)	0.46 (0.56)
L3 (score) ^¥^	0.83 (0.5–1.25)	0.83 (0.67–1.33)	0.75 (0.5–1.33)	0.83 (0.67–1.33)	1.06 (0.57)	1.13 (0.40)	0.08 (0.81)
L4 (score) ^¥^	0.83 (0.33–1.25)	0.83 (0.33–1.33)	0.83 (0.33–1.33)	0.83 (0.33–1.17)	0.66 (0.57)	0.03 (0.87)	1.38 (0.49)
L5 (score) ^¥^	0.27 (0–0.67)	0.0 (0–0.33)	0.17 (0–0.67)	0.17 (0–0.33)	1.46 (0.36)	0.27 (0.67)	0.31 (0.64)

BMI: body mass index, BMI-Z: body mass index standardized according to Croatian normative values for age and sex, WHR—waist to hip ratio, WC—waist circumference, L1 to L5—physical activity levels obtained by Preschool-age Children’s Physical Activity Questionnaire (L1—stationary with no movement, L2—stationary with limb or trunk movement, L3—slow activity, L4—medium activity, L5—fast-paced activity), * denotes variables with Means ± Standard Deviations reported; ^¥^ variables with Medians (25th–75th Percentile) reported, MD—Mediterranean diet, L/A-MD—low and average MD adherence, H/MD—high MD adherence.

**Table 2 ijerph-16-03237-t002:** Mixed effects logistic regression models (with kindergarten as the random factor) indicating factors associated with MD-adherence in preschoolers (Model 1—nonadjusted model; Model 2—adjusted for sex).

	Model 1			Model 2		
	OR *	95% CI	*p*	OR *	95% CI	*p*
BMI-Z (z-score)	1.52	0.99–2.24	0.06	1.37	0.86–2.00	0.17
WHR (ratio)	0.42	0.09–1.95	0.27	0.49	0.12–2.13	0.34
WC (cm)	0.90	0.81–0.97	0.01	0.95	0.90–1.01	0.06
L1 (score)	0.65	0.48–0.98	0.03	0.64	0.45–0.91	0.02
L2 (score)	1.29	0.83–1.98	0.24	1.45	0.92–2.28	0.12
L3 (score)	1.55	0.97–2.50	0.07	1.54	0.96–2.4	0.08
L4 (score)	0.94	0.60–1.53	0.73	0.90	0.58–1.46	0.65
L5 (score)	0.6	0.27–1.31	0.21	0.74	0.35–1.52	0.40

*: per unit of the measure; BMI-Z: body mass index standardized according to Croatian normative values for age and sex, WHR—waist to hip ratio, WC—waist circumference, L1 to L5—physical activity levels obtained by Preschool-age Children’s Physical Activity Questionnaire (L1—stationary with no movement, L2—stationary with limb or trunk movement, L3—slow activity, L4—medium activity, L5—fast-paced activity).

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
