# Peer review of "Physical Activity, Body Mass, and Adherence to the Mediterranean Diet in Preschool Children: A Cross-Sectional Analysis in the Split-Dalmatia County (Croatia)"

_ijerph, 2019, doi:10.3390/ijerph16183237_

Round 1

Reviewer 1 Report

While some of the issues from the previous version of this manuscript have been addressed, some of the more substantial ones have not yet been addressed, or at least not fully addressed, in my opinion, in this version of the manuscript. At a minimum 1) the statistical analyses need to incorporate the clustering within kindergartens, and possibly families, or a convincing argument against this needs to be provided, 2) appropriate model diagnostics need to be presented in the statistical methods and the apparent issues in some of the tables where these assumptions are violated addressed, including for both descriptive and inferential statistics, 3) potential confounders need to be added to the models, 4) the multiple comparison issue needs to be addressed, and 5) effect modification by sex (which is listed as part of the study hypothesis and where evidence for this is claimed in the discussion) needs to be properly tested using an interaction.

There is still work to be done on the language in the manuscript, particularly in the edited/new sections (e.g. Lines 32–36).

I will first discuss some new comments, and then revisit previous comments.

The study hypotheses (Lines 94–95) are important, but it is not clear in the introduction how these were motivated. While you note that there is a lack of studies looking at the association between physical activity, diet, and body composition in Croatia and south-eastern Europe, and personally I would be careful about making absolute statements such as this, how does the literature guide you to the hypothesis that physical activity and Mediterranean diet would be associated in this population, and why do you feel that there would be effect modification by sex?

The sample size section (Lines 120–123) is also important, but I cannot replicate the calculation as it is not clear what effect and what size for that effect are being used for the calculation (nor is the power stated).

P-values need to be added to Lines 188–191 and/or Figure 2.

While forest plots can be useful, I think that readers will also want ORs, 95% CIs, and p-values to go with all of the results in Figures 3–5, not just those mentioned in the text. Note also that the explanation on Lines 249–250 doesn’t reflect that diet is the dependent variable in the model.

In terms of comments also raised previously, note that reporting p-values as “<0.05” (e.g. Lines 217–220) is not particularly useful for readers and actual p-values (with “<0.001” for very small values), along with 95% CIs in most cases, need to be reported alongside effects in the text when appropriate and in all tables in all cases. This will also help in the discussion when interpreting non-statistically significant results. More interpretation of the clinical or practical significance of effects (and potential effects as appropriate) would greatly strengthen the discussion.

Means and  SDs are reported in Table 1, but the values for some variables makes it very clear that there was skew in both boys and girls (including all Pre-PAQ items). This is not to say that other measures in Table 1 are not also skewed, merely that, if present, their skew is not obvious just from looking at the table. A discussion of model diagnostics remains needed in the statistical methods (Lines 170–174), and this should cover the ANOVAs, correlations, and logistic regression. As noted previously, there is some suggestion from looking at the tables that homoscedasticity might also be a problem in a few models along with non-normality.

There are a very large number of tests performed (Table 1 includes 20 tests, 10 for levels of Mediterranean diet adherence, Table 2 another 81 tests, 18 involving diet; Figure 3 implies 10 tests, including one for sex; Figure 4 another 9; and 9 more in Figure 5). This appears to total well over 100 tests and so five or more false positive results under a global null hypothesis would not be surprising. Given that the tests are not independent, this is a simplification, but it does illustrate that with so many tests performed, there is a high risk that several, even most, of the reported associations are spurious. Could the authors please address this matter in their statistical methods and discussion?

Given the interest in stratifying results by boys and girls (e.g. Tables 2 and Figures 4 and 5), the study hypothesis (Line 95), and claims in the discussion (Lines 265–266), a regression-based approach, that also adjusts for possible confounders, would have been much more useful in allowing discussion around effect modification for the correlations as well as the logistic regression models. You claim that there is evidence of effect modification on Lines 265–266, yet at no point do you test for this. Since no possible confounders are accounted for, this also considerably weakens the interpretation of results. It seems unlikely that no additional information about the children, parents, and families that might address this, at least in part, was collected.

If many of the children were recruited via kindergartens (as Lines 294–299 suggest and you mention again this shared factor on Lines 335–337), this would potentially introduce cluster effects that would require a different analysis approach. How were multiple children from the same families treated in the analyses? They also cannot be treated as independent in this case (sharing genetic and/or environmental factors).

While you allude to confounding as a possible explanation of the results (Lines 371–373), I think that confounding due to socio-economic status and parental lifestyle measures are overwhelming likely to explain your findings here. While I could be incorrect in this, considering whether the explanations you propose or simply confounding are more likely cannot be done without modelling these confounders. At the moment, a short list of potential confounder is provided (Line 373), but this would not, for example, help a researcher in planning a study that better addressed this crucial limitation of the present work.

Author Response

Please find responses in the attached file.

Thank you!

Reviewer 2 Report

Revisions have been addressed. 

The manuscript has improved its reliability 

Author Response

Thank you for recognizing the quality in our manuscript. 

Authors

Round 2

Reviewer 1 Report

Thank you for your revisions, which have addressed some of my previous comments. However, there still remain some aspects of those comments that I feel are not yet fully addressed (specifically major points 1, 2 [partially], and 4 [partially] from my previous review). I also have a few smaller comments further below, one following up on a previous point about the sample size calculation and a few comments about edited/added text. I think that addressing the clustering issue, and I have tried to be clearer about what I mean here, and these other comments should not be too difficult given all of the other improvements that the authors have made to their manuscript, which is much improved from last time.

The clustering within kindergartens is not about state-financed versus private, but rather that each kindergarten will have children with more similar diets within it compared to children between kindergartens. This might reflect selection of kindergartens by parents (with social/geographical clustering) and kindergarten practices (with particular staff being more or less enthusiastic about aspects of diet, etc.). Either way, it will mean that data on children’s diets are not independent within the same kindergarten. This is especially important given your comment that “The children we observed were in the last year of preschool education; therefore, they spent most of their day in a kindergarten and were exposed to healthy dietary habits. At this age, this exposure probably translates into their nutritional habits at home.” (Lines 312–314) For example, if there are four children, A1 and A2 who both attend the same kindergarten and B1 and B2 who both attend another kindergarten, it seems plausible that the KIDMED scores (and levels of PA, etc.) for A1 and A2 will be more similar on average than A1 and B1 or than A1 and B2, etc.; similarly, it seems plausible that B1 and B2 will have more similar scores on average than A2 and B2, for example. This possibility of clustering means that standard statistical analyses that assume independent observations are not appropriate. It might be that the cluster effects are minimal, but this could only be known by modelling them or by having a very strong a priori argument (I’ll note that including litter as a cluster effect is becoming increasingly common in animal studies even given that inbreeding and standardized protocols aim to minimize this non-independence of animals within litters), and once this is done, the models that include the clustering are still preferred irrespective of the degree of clustering. Clustering could be accommodated using a random effect in a mixed model (linear mixed regression models and mixed logistic regression models), survey-adjusted regression, GEEs, or robust clustered standard errors. The argument that clustering within families is negligible similarly depends on the number of twins and the degree of similarity. I completely accept that this might be an unusual occurrence, but the strictly correct approach would still be to accommodate this, either through an additional family-level random effect for a mixed model approach or by excluding one of the twins. If they are rare, this second option will involve losing very little data. I will stress that none of this is (necessarily) because the clustering is interesting in itself, but rather the mathematics behind the standard errors and consequently confidence intervals and p-values are only valid for independent observations. If the authors found that there were no twins in the study, this would remove the second possible source of clustering.

While log-transformations might help in addressing positive skew or mean-proportional variation in the statistical models, the reporting of means and SDs are still problematic when there is so much skew in some measures. Replacing means and SDs with geometric means and geometric SDs might help in such cases, or you could report medians and IQRs. Reporting means and SDs is not useful when they imply impossible values under the assumption of an approximately normal distribution (unless another distribution has been specified). For example, the L5 score for boys in Table 1 shows a mean of 0.39 and a SD of 0.49. These are easiest to interpret assuming a normal distribution, where 95% of observations should fall within 1.96 SD of the mean (-0.57 to 1.35 in this case). As -0.57 is clearly an impossible value, the reader cannot use the mean and SD to mentally picture the distribution of the data, beyond knowing that the median must be lower than the mean of 0.39 with strong positive skew in the data. It would be clearer to report geometric means and geometric SDs, as long as there are no zeros although a small constant can be added to allow the calculations in that case, or medians and IQRs (or 25th and 75th percentiles are sometimes used) instead.

There is now a mention of model diagnostics for logistic regression, thank you, but not for the ANOVAs. As I noted previously: “A discussion of model diagnostics remains needed in the statistical methods (Lines 170–174), and this should cover the ANOVAs, correlations, and logistic regression.” If log-transformations were used with this in mind, this would require diagnostics detecting issues with model residuals, the transformation being used, and the diagnostics being used to check for issues with model residuals using the new version of the dependent variable. All of these needs to be (briefly) discussed in the statistical methods in sufficient detail for the reader to feel that the results are valid. This might not involve much work on the authors’ part if this is what they have already done, with a couple of sentences explaining this being added to the methods.

The issue of multiplicity does not invalidate your findings, but it should lead to readers being cautious about interpreting your findings too strongly until they have been replicated. I would suggest a limitation around Line 403 (or perhaps 408) saying something along the lines that many statistical tests were performed, so some of the statistically significant findings are potentially type I errors and that the associations identified here will need to replicated in future studies.

Further comments include:

The sample size section (Lines 142–145) remains un-replicable as it stands. There is not enough information here for the reader to check the sample size as the detectable effect is not described. As noted last time: “I cannot replicate the calculation as it is not clear what effect and what size for that effect are being used for the calculation”. When you performed this calculation, there must have been at least one other value used in the formula or provided to the software.

I’m slightly confused about Line 233: “The sex x MD interaction effect was significant for L1 (F-test: 4.29, p = 0.04) (Table 1).” There are no values of 4.29 in the table, and the closest match appears to be for gender (and you need to standardized references to either gender [the social construct] or sex [the biological state] throughout the manuscript) where the main effect has F=4.28 and p=0.04. The interaction p-value for this row is 0.09, non-statistically significant. Am I misreading the table and/or text here?

In Table 2, ORs are presented for several independent variables. The units of these need to be made explicit. I am assuming, but it is not definite, that these are per single point for each scale? For WC, this is presumably “per cm”, for example? This should also be made clear on Lines 37–38 in the abstract and Lines 247–267 in the results, where the per unit should be made clear for the first instance of each independent variable.

Line 263’s “(OR: 1.74; 95% CI: 1.04-1.89, p = 0.03)” cannot be correct (the OR is incompatible with the CI limits: you can check this using the square of the OR=lower CI x Upper CI). From Figure 3, it looks as if it might be an upper CI limit of 2.89 instead. It would be worth checking all other results reported here for possible typos also.

Author Response

RESPONSES

Thank you for your revisions, which have addressed some of my previous comments. However, there still remain some aspects of those comments that I feel are not yet fully addressed (specifically major points 1, 2 [partially], and 4 [partially] from my previous review). I also have a few smaller comments further below, one following up on a previous point about the sample size calculation and a few comments about edited/added text. I think that addressing the clustering issue, and I have tried to be clearer about what I mean here, and these other comments should not be too difficult given all of the other improvements that the authors have made to their manuscript, which is much improved from last time.

RESPONSE: Thank you for recognizing our efforts. Also, we must thank you once again for elaborated and constructive review(s). We tried to specifically follow it and amended the manuscript accordingly. Please see bellow how did we modify the manuscript with regard to your comments.

The clustering within kindergartens is not about state-financed versus private, but rather that each kindergarten will have children with more similar diets within it compared to children between kindergartens. This might reflect selection of kindergartens by parents (with social/geographical clustering) and kindergarten practices (with particular staff being more or less enthusiastic about aspects of diet, etc.). Either way, it will mean that data on children’s diets are not independent within the same kindergarten. This is especially important given your comment that “The children we observed were in the last year of preschool education; therefore, they spent most of their day in a kindergarten and were exposed to healthy dietary habits. At this age, this exposure probably translates into their nutritional habits at home.” (Lines 312–314) For example, if there are four children, A1 and A2 who both attend the same kindergarten and B1 and B2 who both attend another kindergarten, it seems plausible that the KIDMED scores (and levels of PA, etc.) for A1 and A2 will be more similar on average than A1 and B1 or than A1 and B2, etc.; similarly, it seems plausible that B1 and B2 will have more similar scores on average than A2 and B2, for example. This possibility of clustering means that standard statistical analyses that assume independent observations are not appropriate. It might be that the cluster effects are minimal, but this could only be known by modelling them or by having a very strong a priori argument (I’ll note that including litter as a cluster effect is becoming increasingly common in animal studies even given that inbreeding and standardized protocols aim to minimize this non-independence of animals within litters), and once this is done, the models that include the clustering are still preferred irrespective of the degree of clustering. Clustering could be accommodated using a random effect in a mixed model (linear mixed regression models and mixed logistic regression models), survey-adjusted regression, GEEs, or robust clustered standard errors. The argument that clustering within families is negligible similarly depends on the number of twins and the degree of similarity. I completely accept that this might be an unusual occurrence, but the strictly correct approach would still be to accommodate this, either through an additional family-level random effect for a mixed model approach or by excluding one of the twins. If they are rare, this second option will involve losing very little data. I will stress that none of this is (necessarily) because the clustering is interesting in itself, but rather the mathematics behind the standard errors and consequently confidence intervals and p-values are only valid for independent observations. If the authors found that there were no twins in the study, this would remove the second possible source of clustering.

RESPONSE: Thank you for your clear elaboration, it helped a lot. We truly appreciate it. Namely, in the last version of the manuscript we have followed the findings of the previous studies done in the field where authors frequently evidenced problem of “state-financed vs. private-kindergartens”. However, it is absolutely clear that the “cluster effect” for “kindergartens itself” may appear as you specified (i.e. tendency of parents, enthusiasm of the employees). Therefore, in this version of the manuscript we tried to resolve this issue. In short, first we calculated the intracluster coefficient for “kindergartens as clusters”, and found (as you hypothesized) significant differences between kindergartens in MD. However, it appeared that these differences are actually a result of the differences between “coastal” and “inland” region since additional calculation of the intracluster coefficient separately for “inland” and “coastal” kindergartens did not reveal differences among kindergartens within each region. As a result, since in previous version of the manuscript we included “region” as covariate in logistic regression, herein we did not perform additional analyses. We know that the “lack of differences” between kindergartens in each region may seem surprising, but we must note that the region we observed herein is actually small (less than 600,000 residents in total; note that population of Croatia is less than 4 million), and that practically all state-financed kindergartens follow maybe not “equal”, but almost certainly “standardized” (i) nutritional- and (ii) daily activity-plans. Therefore (we are speaking from our own experience as parents) most parents actually choose kindergarten simply because of some “individual” circumstances which are not unique (i.e. kindergartens closer to grandparents, closer to parental job facility), which altogether actually result in “randomization” of studied effects.

Meanwhile, and as you specified in your comment, we checked the data and found “two pairs of twins”, and randomly excluded one twin from each pair, and re-calculated the analyses once again. Generally, results of the analyses did not change (although small differences are evident for some numerical parameters).  The results of intracluster analyses are presented in the Methods section, and text reads: “In order to test possible similarity in MD and PA-levels among children attending same kindergartens we calculated intracluster correlation (IC) with kindegartens as clusters [26,27]. In brief, the IC for PA-levels showed appropriate within-kindergarten variance (IC: 0.03 to 0.06), but the IC for MD-score showed reduced variability of results in certain kindergartens (IC: 0.14). However, when IC was calculated separately for inland and coastal kindergartens, the IC values dropped down to appropriate levels (IC: 0.04 and 0.07 for inland and coastal region, respectively), which actually indicated differences between studied regions. As a result, logistic regression analyses (please see Statistics for details) included “region” (coast vs. inland) as covariate.” (Please see last paragraph of the Participants subsection)

While log-transformations might help in addressing positive skew or mean-proportional variation in the statistical models, the reporting of means and SDs are still problematic when there is so much skew in some measures. Replacing means and SDs with geometric means and geometric SDs might help in such cases, or you could report medians and IQRs. Reporting means and SDs is not useful when they imply impossible values under the assumption of an approximately normal distribution (unless another distribution has been specified). For example, the L5 score for boys in Table 1 shows a mean of 0.39 and a SD of 0.49. These are easiest to interpret assuming a normal distribution, where 95% of observations should fall within 1.96 SD of the mean (-0.57 to 1.35 in this case). As -0.57 is clearly an impossible value, the reader cannot use the mean and SD to mentally picture the distribution of the data, beyond knowing that the median must be lower than the mean of 0.39 with strong positive skew in the data. It would be clearer to report geometric means and geometric SDs, as long as there are no zeros although a small constant can be added to allow the calculations in that case, or medians and IQRs (or 25th and 75th percentiles are sometimes used) instead.

RESPONSE: Thank you for your comment. In this version of the manuscript we reported Medians and 25-75 percentiles) for PA levels. Actually, there were some numerical “zeros” for these variables so we decided not to add numerical constant but rather to present medians and 25-75th percentiles as you suggested (for details please see Table 1). We must mention that means and SDs were calculated for anthropometrics. Accordingly, we changed the presentation of the Statistical analyses, and text now reads: “The descriptive statistics included means and standard deviations (for normally distributed parametric variables), medians and 25th to 75th percentiles (for PA-levels), and percentages (for categorical variables).” (Please see first part of Statistical subsection).  

There is now a mention of model diagnostics for logistic regression, thank you, but not for the ANOVAs. As I noted previously: “A discussion of model diagnostics remains needed in the statistical methods (Lines 170–174), and this should cover the ANOVAs, correlations, and logistic regression.” If log-transformations were used with this in mind, this would require diagnostics detecting issues with model residuals, the transformation being used, and the diagnostics being used to check for issues with model residuals using the new version of the dependent variable. All of these needs to be (briefly) discussed in the statistical methods in sufficient detail for the reader to feel that the results are valid. This might not involve much work on the authors’ part if this is what they have already done, with a couple of sentences explaining this being added to the methods.

RESPONSE: Indeed, in the previous version of the manuscript we did not specify the validation of the ANOVA calculation. Specifically, ANOVA model diagnostics included the analysis of normality for the distribution(s) of residuals. It is now specified in the Statistics subsection. Text reads: “The distribution of residuals was checked by Kolmogorov Smirnov test, and were found to be normally distributed for all dependent variables except for BMI, indicating that results for most of the ANOVA calculations were reliable and readily interpretable.” (please see 2nd paragraph of the Statistics)

The issue of multiplicity does not invalidate your findings, but it should lead to readers being cautious about interpreting your findings too strongly until they have been replicated. I would suggest a limitation around Line 403 (or perhaps 408) saying something along the lines that many statistical tests were performed, so some of the statistically significant findings are potentially type I errors and that the associations identified here will need to replicated in future studies.

RESPONSE: Thank you for this comment. As you suggested we included text about multiplicity as a certain limitation of our study. Text reads: “Finally, we must mention that this study included many statistical tests, and therefore, there is a possibility that some statistically significant findings are potentially a result of “type I errors”. As a result, the findings from this study should be replicated in future investigations.” (Please see end of first paragraph of Limitations subsection)

 Further comments include:

 The sample size section (Lines 142–145) remains un-replicable as it stands. There is not enough information here for the reader to check the sample size as the detectable effect is not described. As noted last time: “I cannot replicate the calculation as it is not clear what effect and what size for that effect are being used for the calculation”. When you performed this calculation, there must have been at least one other value used in the formula or provided to the software.

RESPONSE: According to your comment we included all parameters used in calculation of the sample size, which was done using the freely available online calculator provided by University of Zagreb, Faculty of Medicine (http://web.mef.unizg.hr/if/alati/racunala/skripte/velicina_ap.htm; ). Text reads: Based on (i) an estimated 20% incidence of a poor KIDMED score [24], (ii) a population sample of approximately 2500 preschoolers in the studied region (Split-Dalmatia County), (iii) absolute error of 0.05, (iii) reliability of 0.95, and (iv) statistical power of 0.80, the required sample size for this investigation was calculated to be 225 participant, as calculated by online software provided by University of Zagreb, Faculty of Medicine (Zagreb, Croatia) [25]. (Please see last part of Participants subsection)

I’m slightly confused about Line 233: “The sex x MD interaction effect was significant for L1 (F-test: 4.29, p = 0.04) (Table 1).” There are no values of 4.29 in the table, and the closest match appears to be for gender (and you need to standardized references to either gender [the social construct] or sex [the biological state] throughout the manuscript) where the main effect has F=4.28 and p=0.04. The interaction p-value for this row is 0.09, non-statistically significant. Am I misreading the table and/or text here?

RESPONSE: Thank you for noticing this mistake. It is corrected now (no significant Sex x MD interaction). Also, the term “gender” is replaced with “sex” throughout the manuscript since parents actually declared biological state of their children.

In Table 2, ORs are presented for several independent variables. The units of these need to be made explicit. I am assuming, but it is not definite, that these are per single point for each scale? For WC, this is presumably “per cm”, for example? This should also be made clear on Lines 37–38 in the abstract and Lines 247–267 in the results, where the per unit should be made clear for the first instance of each independent variable.

RESPONSE: The units for all variables are presented in Tables, Results and in the Abstract. Thank you.

Line 263’s “(OR: 1.74; 95% CI: 1.04-1.89, p = 0.03)” cannot be correct (the OR is incompatible with the CI limits: you can check this using the square of the OR=lower CI x Upper CI). From Figure 3, it looks as if it might be an upper CI limit of 2.89 instead. It would be worth checking all other results reported here for possible typos also.

RESPONSE: Thank you, this was a typing mistake (the correct 95%CI is as you supposed 1.04-2.89). We checked other numbers for typos as well.

Thank you once again for such elaborated and detailed review and comments. We must say that we learned a lot.

Staying at your disposal.

Authors

Round 3

Reviewer 1 Report

Thank you for your revisions. We are now down to just three comments on my part, two of which I think are fairly trivial and just one that is a bit more substantial.

First, I appreciate your investigation of the clustering and for addressing the very minor issue with twin participants. While you have looked at ICCs within kindergartens for MD, note that it is these conditional on the model covariates that affects the standard errors (reflecting what you did by stratifying by region but pooled over the entire sample). It is mathematically possible for high ICCs to disappear after adjusting for covariates, and also for ICCs very close to zero to become high after adjusting for covariates. Even if the ICCs were close to zero (and yours are not that close) after stratifying by region, this still wouldn’t justify not accounting for the clustering as your models include another variable (sex) also. More importantly, even low ICCs can still affect the results substantially. The only appropriate ICC for ignoring clustering would be 0! If the degree of clustering (the conditional ICC) was extremely low (I’d say this is ICC<0.01) and cluster sizes small (say < 10), accounting for this clustering is still the correct thing to do and in these cases will have little effect on the standard errors (they would be around 4% smaller than they should be), only slightly shifting the p-values and CI limits. The clustered analyses should be presented irrespective of the degree of clustering. With your strata-specific ICCs (0.04 and 0.07 for the two regions), if each kindergarten contributes 20 participants (this wasn’t clear to me from the manuscript), this would still produce design effects of 1.76 and 2.33, meaning that your effective sample size is actually around one half of what it appears to be and so the standard errors would be around 40% smaller than they should be. The design effects would be larger if there are fewer kindergartens and smaller if there are more, but even with only 10 children on average per kindergarten, you would still have design effects of 1.36 and 1.63, or standard errors about 20% too small.

I’m sorry, but mixed logistic regression (or GEEs or robust clustered standard errors) are still needed instead of logistic regression (Line 215) and similarly linear mixed models (or GEEs or robust clustered standard errors) are still needed instead of ANOVAs (Line 210). The Chi-squared test (Line 209) can be replaced with a mixed logistic regression model (or similar) or a survey-adjusted Chi-squared test used.

For the diagnostics, there are some nomenclature changes I’d suggest: Line 204 “nonuniformity” => “non-normality” and/or “heteroscedasticity” (depending on what you meant here). And was “All data” (Line 204) transformed or just those where model residuals were skewed (from Lines 211–214)? Assuming the latter, this first sentence (Line 204) would be more naturally placed around Line 214.

For the sample size, based on Google translate, this page is for survey margins of error (as you might see with opinion polls). This would mean that your study was powered to estimate the proportion of poor KIDMET scores, assuming this is 20% in a population of 2500, with the 95% confidence interval no wider than ±0.05, i.e. sufficiently large so that an observed result of 20% for poor KIDMET scores would have a 95% CI of 15%–25%. This is slightly different to what is written in the manuscript on Lines 143–147. The “absolute error” here appears to be the confidence interval half-width and “reliability” here appears to be talking about the confidence interval level—the latter of these may well be a translation issue as Google translate also used this word in the English translation—and note that there is no “statistical power of 0.80” here. Power would be used if you wanted a certain chance to detect a particular difference between groups or an association between variables.

You can replicate the calculations in Stata using the downloadable svysampsi program:

. svysampsi 2500, prop(0.2) moe(5) level(95)

Estimated sample size needed to survey, assuming the following:

   Population size: 2500

   Proportion of sample with the expected outcome: 0.20

   Margin of error: +/-  5.0 %

   Confidence level: 95.0 %

Estimated required sample size:

       n = 224

Or in R using the PracTools package (there are many other options as far as packages in R go):

> nPropMoe(moe.sw=1,e=0.05,alpha=0.05,pU=0.20,N=2500)

[1] 223.9221

Both giving n=224, which is close enough to your n=225 to be due to rounding/implementation differences. Using https://goodcalculators.com/margin-of-error-calculator/ will also show that n=224 gives a “Margin of Error: ±4.999%” (with n=223’s MOE being > 5%).

Author Response

RESPONSE TO REVIEW

Dear Sirs

Once again, we must express our sincere gratitude for Reviewer’s elaborated comments and suggestions. In general, we accepted all comments and suggestions and amended the manuscript accordingly.

Most important changes:

As reviewer suggested, in this version of the manuscript we calculated mixed effect logistic regression (with kindergartens as fixed factor), and results are presented accordingly Also, mixed model of analysis of variance was calculated and presented Amended tables and Figure (Figure 3) are included

Please find bellow for specific amendments (indicated in RESPONSES). The changes are evidenced in the “highlighted” parts of the text.

Staying at your disposal

Authors

REVIEW:

Thank you for your revisions. We are now down to just three comments on my part, two of which I think are fairly trivial and just one that is a bit more substantial.

First, I appreciate your investigation of the clustering and for addressing the very minor issue with twin participants. While you have looked at ICCs within kindergartens for MD, note that it is these conditional on the model covariates that affects the standard errors (reflecting what you did by stratifying by region but pooled over the entire sample). It is mathematically possible for high ICCs to disappear after adjusting for covariates, and also for ICCs very close to zero to become high after adjusting for covariates. Even if the ICCs were close to zero (and yours are not that close) after stratifying by region, this still wouldn’t justify not accounting for the clustering as your models include another variable (sex) also. More importantly, even low ICCs can still affect the results substantially. The only appropriate ICC for ignoring clustering would be 0! If the degree of clustering (the conditional ICC) was extremely low (I’d say this is ICC<0.01) and cluster sizes small (say < 10), accounting for this clustering is still the correct thing to do and in these cases will have little effect on the standard errors (they would be around 4% smaller than they should be), only slightly shifting the p-values and CI limits. The clustered analyses should be presented irrespective of the degree of clustering. With your strata-specific ICCs (0.04 and 0.07 for the two regions), if each kindergarten contributes 20 participants (this wasn’t clear to me from the manuscript), this would still produce design effects of 1.76 and 2.33, meaning that your effective sample size is actually around one half of what it appears to be and so the standard errors would be around 40% smaller than they should be. The design effects would be larger if there are fewer kindergartens and smaller if there are more, but even with only 10 children on average per kindergarten, you would still have design effects of 1.36 and 1.63, or standard errors about 20% too small.

I’m sorry, but mixed logistic regression (or GEEs or robust clustered standard errors) are still needed instead of logistic regression (Line 215) and similarly linear mixed models (or GEEs or robust clustered standard errors) are still needed instead of ANOVAs (Line 210). The Chi-squared test (Line 209) can be replaced with a mixed logistic regression model (or similar) or a survey-adjusted Chi-squared test used.

RESPONSE: First of all, thank you for such profound and detailed explanation. Actually, we had no doubt that your observation was correct, but until now, we tried to deal with “the issue” by different approaches. In this version we calculated mixed model logistic regression, while including kindergartens as fixed effect, as you suggested. Specifically, it changed some coefficients, and consequently the “influence” of L3 on MD adherence is not significant. As a result, some parts of the text are amended as well. Further, as you suggested we calculated linear mixed model ANOVA as well (with kindergartens as fixed effect), and amended Tables and text of the Results accordingly. With regard to changes of logistic regression calculation the text now reads:

In subsection Statistics:

The mixed model logistic regression (with kindergarten as random factor) was applied to demonstrate the correlates of MD adherence (observed as binomial criterion: LA/MD [coded as “1”] vs. H-MD [coded as “2], as suggested previously [34]). The regression calculations included nonadjusted model (Model 1) model adjusted for sex (Model 2), and model adjusted for sex and location (coast-inland).Goodness of fit was evaluated by the Hosmer-Lemeshow test (a statistically significant result indicates that the model does not adequately fit the data).

In subsection Results:

Table 2; Figure 3 and text: The mixed effect logistic regression Model 1 (with kindergarten as random factor) with MD adherence as a criterion showed a significant correlation for WC (OR: 0.90, 95% CI. 0.81-0.97, p < 0.01), and L1 (OR: 0.65 95% CI: 0.48-0.98, p = 0.03). Specifically, H-MD adherence was evidenced in children with a low WC and low sedentary activity. When adjusted for sex (Model 2), significant predictor of MD adherence was L1 (OR: 0.64, 95% CI: 0.45-0.91, p = 0.02) (Table 2). The regression Model 3 (adjusted for sex and location) identified L1 (OR per score: 0.65; 95% CI: 0.44-0.91, p = 0.02) as significant correlates of MD adherence, with a higher likelihood for H-MD adherence in children with low scores on L1. The remaining variables were not significantly correlated with MD (OR [95% CI]; BMI-Z: 1.13 [0.74-1-76], WHR: 0.47 [0.09-2.30], WC: 0.94 [0.83-1.05], L2: 1.46 [0.90-2.34], L3: 1.76 [0.97-2.92], L4: 0.90 [0.56-1.48], and L5: 0.84 [0.36-1.86]) (Figure 3). The Hosmer–Lemeshow test results (Chi square = 6.31, 7.04, 9.52, p = 0.62, 0.55, 0.30 for Model 1, Model 2, and Model 3, respectively) indicate that the goodness of fit was satisfactory.

Also, ANOVA model was changed and now includes fixed effect of kindergarten, as you suggested, with following amendments in text:

In subsection Statistics:

The linear mixed model (with kindergartens as random factor) of analysis of variance (ANOVA), was calculated to identify main effects (sex, MD adherence) and interactions (sex x MD adherence) in the studied variables. The distribution of residuals was checked by Kolmogorov Smirnov test, and were found to be normally distributed for all dependent variables except for BMI, indicating that results for most of the ANOVA calculations were reliable and readily interpretable.

The Results

Table 1 Text: “The ANOVA main effects of MD adherence were significant for L1 (F-test: 22.23, p = 0.03) with lower sedentary activity in children with good MD adherence. (Table 1).”

For the diagnostics, there are some nomenclature changes I’d suggest: Line 204 “nonuniformity” => “non-normality” and/or “heteroscedasticity” (depending on what you meant here). And was “All data” (Line 204) transformed or just those where model residuals were skewed (from Lines 211–214)? Assuming the latter, this first sentence (Line 204) would be more naturally placed around Line 214.

RESPONSE: Thank you. Amended as suggested. However, log-transformations was done for all variables. Text reads: “All data were log-transformed to reduce the non-normality of errors [33].” (please see first sentence of Statistics)

For the sample size, based on Google translate, this page is for survey margins of error (as you might see with opinion polls). This would mean that your study was powered to estimate the proportion of poor KIDMET scores, assuming this is 20% in a population of 2500, with the 95% confidence interval no wider than ±0.05, i.e. sufficiently large so that an observed result of 20% for poor KIDMET scores would have a 95% CI of 15%–25%. This is slightly different to what is written in the manuscript on Lines 143–147. The “absolute error” here appears to be the confidence interval half-width and “reliability” here appears to be talking about the confidence interval level—the latter of these may well be a translation issue as Google translate also used this word in the English translation—and note that there is no “statistical power of 0.80” here. Power would be used if you wanted a certain chance to detect a particular difference between groups or an association between variables.

You can replicate the calculations in Stata using the downloadable svysampsi program: svysampsi 2500, prop(0.2) moe(5) level(95); Estimated sample size needed to survey, assuming the following:    Population size: 2500; Proportion of sample with the expected outcome: 0.20;    Margin of error: +/-  5.0 %; Confidence level: 95.0 %; stimated required sample size: n = 224

Or in R using the PracTools package (there are many other options as far as packages in R go): > nPropMoe(moe.sw=1,e=0.05,alpha=0.05,pU=0.20,N=2500); [1] 223.9221

Both giving n=224, which is close enough to your n=225 to be due to rounding/implementation differences. Using https://goodcalculators.com/margin-of-error-calculator/ will also show that n=224 gives a “Margin of Error: ±4.999%” (with n=223’s MOE being > 5%).

RESPONSE: Thank you. We recalculated the sample size at proposed web page (https://goodcalculators.com/sample-size-calculator/) and text is amended accoridnlgy. It now reads: „Based on (i) an estimated 20% incidence of a poor KIDMED score [24], (ii) a population sample of approximately 2500 preschoolers in the studied region (Split-Dalmatia County), (iii) margin of error of 0.05, and (iii) confidence level of 0.95, the required sample size for this investigation was calculated to be 224 participants.“ (please see last paragraph of Participants subsection).

Thank you once again!

Authors

Round 4

Reviewer 1 Report

Thank you for your revisions and responses to my queries. Aside from some minor suggestions to improve clarity and flow, and a note that a Chi-squared test could have been replaced with logistic regression to allow modelling the clustering within kindergartens, I have no further comments.

Note that the analyses described on Line 196 (“The MD adherence status for boys and girls was compared by the Chi-square test.”) could also be performed using mixed effects logistic regression, but I’ll leave this as a suggestion only.

The units on Line 38 are appreciated, but perhaps “per point” rather than “per score”. The same applies to Line 251. Note that the units are not provided on Lines 247, 248, 250, 254–255. I also suggest “per point” rather than “score” in Table 2, as well as “per z-score”, “per unit of the ratio”, and “per cm”. Or you could use “point” rather than “score” in Table 2 and annotate the table that ORs are per unit of the measure as indicated in parentheses.

The “correlated” on Line 38 would normally be described as “associated”. This is particularly the case for non-linear models such as logistic regression, but this would be the preferred term in all cases that are not actual correlations (Pearson, Spearman, intra-class, etc.). The same applies to Lines 79 (“associations”), 80 (“association”), 81 (“associated”, twice), 86 (“association”), 89 (“association”), 110 (“associated”), 203 (“identify the variables associated with MD adherence”), 247 (“association”), 252 (“as significantly associated with MD adherence”), 254 (“associated”), 260 (“associations with MD adherence”), 271 (“Variables associated with MD adherence”), 333 (“association”), 336 (“association”), 376 (“association”), 387 (“associated”), 388 (“associated”), and 425 (“association”).

The linear mixed model, as described on Lines 196–197, is not an ANOVA and so the words “of analysis of variance (ANOVA),” on Line 197 could simply be deleted. The LMM generalises the concept of a G(eneral)LM, which in turn generalises the concept of an ANOVA, but is not an ANOVA. Related to this, the reference to “ANOVA” on Line 201 is also not correct. You could perhaps replace “ANOVA calculations” with “models” here, but I’m not entirely sure what you mean by “reliable and readily interpretable” and you could also just delete “indicating that results for most of the ANOVA calculations were reliable and readily interpretable.” Again, the word “ANOVA” on Line 231 should be deleted. The heading for Table 1 could be “Linear mixed model” rather than “ANOVA” respectively and the caption “linear mixed models” rather than “analysis of variance (ANOVA)”.

There is some punctuation and words missing in “The regression calculations included nonadjusted model (Model 1) model adjusted for sex (Model 2), and model…” and this could be written (with some word substitutions) as “The regression analyses included an unadjusted model (Model 1), a model adjusted for sex (Model 2), and a model…”

The wording on Lines 246–247 is a little awkward and “…with MD adherence as a criterion showed…” might be clearer as “…with MD adherence as the dependent variable showed…”

The word “evidenced” on Line 248 is non-standard and might read better as “found”. Same for Lines 83,  213, 297, 386, and 425.

On Line 249, you’ll want a “a” before “significant predictor”.

Figure 3 needs to make it clear that this is “per” point, etc. again.

Author Response

RESPONSES

Dear Sir/Madam

Thank you again for your efforts and constructive comments and suggestions. We tried to follow all your comments and amended the manuscript accordingly. Please see below how we responded and where to find specific changes.

Staying at your disposal

Authors

Thank you for your revisions and responses to my queries. Aside from some minor suggestions to improve clarity and flow, and a note that a Chi-squared test could have been replaced with logistic regression to allow modelling the clustering within kindergartens, I have no further comments.

RESPONSE: Thank you once again.

Note that the analyses described on Line 196 (“The MD adherence status for boys and girls was compared by the Chi-square test.”) could also be performed using mixed effects logistic regression, but I’ll leave this as a suggestion only.

RESPONSE: Thank you for this suggestion. We were of the opinion that Chi square calculation would be more „explorative“ and understandable, so we left this analysis as it is.

The units on Line 38 are appreciated, but perhaps “per point” rather than “per score”. The same applies to Line 251. Note that the units are not provided on Lines 247, 248, 250, 254–255. I also suggest “per point” rather than “score” in Table 2, as well as “per z-score”, “per unit of the ratio”, and “per cm”. Or you could use “point” rather than “score” in Table 2 and annotate the table that ORs are per unit of the measure as indicated in parentheses.

RESPONSE: Amended accordingly. In all mentioned cases we noted that ORs were expressed per unit of the specific measure (same with Table 2 – please see Table legend). Thank you.

The “correlated” on Line 38 would normally be described as “associated”. This is particularly the case for non-linear models such as logistic regression, but this would be the preferred term in all cases that are not actual correlations (Pearson, Spearman, intra-class, etc.). The same applies to Lines 79 (“associations”), 80 (“association”), 81 (“associated”, twice), 86 (“association”), 89 (“association”), 110 (“associated”), 203 (“identify the variables associated with MD adherence”), 247 (“association”), 252 (“as significantly associated with MD adherence”), 254 (“associated”), 260 (“associations with MD adherence”), 271 (“Variables associated with MD adherence”), 333 (“association”), 336 (“association”), 376 (“association”), 387 (“associated”), 388 (“associated”), and 425 (“association”).

RESPONSE: Thank you for such detailed instructions. In this version of the manuscript the term „correlation“ is systematically changed into „association“ (i.e. associated). For details please see „track changes version of the manuscript“.

The linear mixed model, as described on Lines 196–197, is not an ANOVA and so the words “of analysis of variance (ANOVA),” on Line 197 could simply be deleted. The LMM generalises the concept of a G(eneral)LM, which in turn generalises the concept of an ANOVA, but is not an ANOVA.

RESPONSE: Thank you. Amended accordingly. Text reads: „The linear mixed model (LMM; with kindergartens as random factor) was calculated to identify main effects (sex, MD adherence) and interactions (sex x MD adherence) in the studied variables. The distribution of residuals was checked by Kolmogorov Smirnov test, and were found to be „ (Please see 2nd paragraph of the 3.3. Statistics)

 Related to this, the reference to “ANOVA” on Line 201 is also not correct. You could perhaps replace “ANOVA calculations” with “models” here, but I’m not entirely sure what you mean by “reliable and readily interpretable” and you could also just delete “indicating that results for most of the ANOVA calculations were reliable and readily interpretable.”

RESPONSE: Amended accordingly. Text reads: „The distribution of residuals was checked by Kolmogorov Smirnov test, and were found to be normally distributed for all dependent variables except for BMI.“ (Please see 2nd paragraph of the 3.3. Statistics). Thank you.

Again, the word “ANOVA” on Line 231 should be deleted. The heading for Table 1 could be “Linear mixed model” rather than “ANOVA” respectively and the caption “linear mixed models” rather than “analysis of variance (ANOVA)”.

RESPONSE: Amended accordingly

There is some punctuation and words missing in “The regression calculations included nonadjusted model (Model 1) model adjusted for sex (Model 2), and model…” and this could be written (with some word substitutions) as “The regression analyses included an unadjusted model (Model 1), a model adjusted for sex (Model 2), and a model…”

RESPONSE: Amended accordingly. Text reads: „The mixed model logistic regression (with kindergarten as random factor) was applied to demonstrate the factors associated with MD adherence (observed as binomial criterion: LA/MD [coded as “1”] vs. H-MD [coded as “2], as suggested previously [34]). The regression analyses included an unadjusted model (Model 1), a model adjusted for sex (Model 2), and a model adjusted for sex and location (coast-inland).“ (Please see 3rd paragraph of the Statistics.

The wording on Lines 246–247 is a little awkward and “…with MD adherence as a criterion showed…” might be clearer as “…with MD adherence as the dependent variable showed…”

RESPONSE: Amended accordingly. Text reads: „The mixed effect logistic regression Model 1 (with kindergarten as random factor) with MD adherence as the dependent variable showed a significant association between WC and criterion (OR per cm: 0.90, 95% CI. 0.81-0.97, p < 0.01), etc.“ please see 1st paragraph after Table 1).

The word “evidenced” on Line 248 is non-standard and might read better as “found”. Same for Lines 83,  213, 297, 386, and 425.

RESPONSE: Amended accordingly. Thank you.

On Line 249, you’ll want a “a” before “significant predictor”.

RESPONSE: Added. Thank you

Figure 3 needs to make it clear that this is “per” point, etc. again.

RESPONSE: Amended accordingly. The wording „per unit of the measure“ is added in the X – title (Please see Figure 3)

Staying at your disposal

Authors

This manuscript is a resubmission of an earlier submission. The following is a list of the peer review reports and author responses from that submission.

Round 1

Reviewer 1 Report

This is a cross-sectional article investigating a possible relationship between adherence to the Mediterranean diet, physical activity and body composition in a cohort of Croatian children (aged 5-6 years). The Article is well-written and interesting. The abstract, introduction and discussion are clear.

However, there are some methodological drawbacks (major reviews) that must be addressed before the publication and minor reviews that should be done to improve the text.

Major reviews:

- No sample size calculation is described. The sample is 256, but it could not be representative of the whole preschool Croatian children population. Please, provide a sample size calculation. 

- BMI is not a reliable tool to evaluate body composition at all; moreover, it is not recommended to assess underweight or overweight in children or adolescents. Indeed, looking at the results, it seems that all children are malnourished (BMI < 16). Authors should use appropriate tools to identify nutritional status in children. I suggest (for its simplicity) to use BMI Z-score, that is BMI corrected for age (available at https://zscore.research.chop.edu/), citing these references: 

1) Must A, Anderson SE. Body mass index in children and adolescents: considerations for population-based applications. Int J Obes (Lond). 2006 Apr;30(4):590-4.

2) Rinninella E, Ruggiero A, Maurizi P, Triarico S, Cintoni M, Mele MC. Clinical tools to assess nutritional risk and malnutrition in hospitalized children and adolescents. Eur Rev Med Pharmacol Sci. 2017 Jun;21(11):2690-2701.

Also, a revision of the title should be done. Since BMI does not indicate body composition, this term in the title should be replaced with "body mass".

Minor reviews:

- line: 82: time consuming ---> time-consuming (hyphenated) 

- paragraph 3.2.  Variables should be divided into three paragraphs: Anthropometric measurements, Level of adherence to MD, Physical exercise level.

- As regards the results, the presentation and accuracy of the results should be improved. Indeed, there are too many results tables and they need some revisions and precisions.

- Figure 1: In the text, Authors say that boys have higher adherence to MD. However, the Figure seems to describe the opposite: girls have an 81% rate of high adherence, whereas boys have 59% (less than girls), with a higher rate of low MD adherence (8% vs 3%). Please, clarify. Moreover, in Figure 1 the results for both sex are missing. 

It would be better to present the results of figure 1 and table 1  into a unique and more accurate table: Characteristics of the study sample by sex and adherence to the MD.

Sex

Adherence to MD

Total (N=)

Girls (N=)

Boys (N=)

P value

Low (N=)

Med (N=)

High (N=)

P value

Mean

SD

Mean

SD

Mean

SD

Mean

SD

Mean

SD

Mean

SD

BMI (unit)

WHR (unit)

WC(unit)

Pre-PAQ (unit)

-Tables 1, 3, 4, 5: Please replace  kg/m**2 with kg/m2

- SD are unclear, please specify the abbreviation "standard deviation" and add the symbol  ±  in front of each SD.

- Regarding all tables, the unit of each variable should be indicated in brackets. Moreover, the abbreviations should be indicated at the bottom of each table.

- The results of the tables 3, 4 and 5 could be laid out in a single table. 

- To accurate the results and consequently the discussion, it could be interesting to study the distribution of the KIDMED items with a positive answer (+1) by gender and BMI. 

Reviewer 2 Report

Lejla Obradovic Salcin et al have analyzed the association between adherence to Mediterranean diet (MD), level of physical activity (PA), and body composition indices in healthy preschool children from southern Croatia. The main limitations of the paper include a very low (n= 260), the non objectively measured physical activity, the body composition indices limited to BMI and waist circumference and a lack of novelty in the statistical analyses. All these limitations contribute to the non-suitability for publication in a high impact journal.

Reviewer 3 Report

This survey of n=260 preschoolers in southern Croatia found negative correlations between BMI and PA in boys and between diet quality and both waist circumference and sedentary activity in girls. No adjustment for confounders was performed, making the results difficult to interpret. While the association between BMI and PA (for boys) and between diet quality and WC (for girls) have causal hypotheses that would potentially explain them, the association between diet quality and sedentary activity seems likely to reflect residual confounding.

The study cannot make strong conclusions (due to the cross-sectional design and lack of consideration of confounders), but the authors are mostly careful to avoid causal language and acknowledge some of the study limitations.

Note that reporting p-values as “<0.05” (e.g. Lines 32 and 22) is not particularly useful for readers and 95% CIs and actual p-values (with “<0.001” for very small values) need to be reported alongside effects in the text when appropriate and in all tables in all cases. This will also help in the discussion when interpreting non-statistically significant results. More interpretation of the clinical or practical significance of effects (and potential effects as appropriate) would greatly strengthen the discussion.

I’m not sure why you didn’t use BMI z-scores given that there are differences between boys and girls and between ages 5 and 6. Could you please justify this choice or use z-scores instead?

The statistical analyses state that normality was examined, but some of the data in Table 1 makes it quite clear that there was skew in both boys and girls (including all Pre-PAQ items). This is not to say that other measures in Table 1 are not also skewed, merely that their skew is not obvious just from looking at the table. The model diagnostics would also need to include checks for homoscedasticity (there is some suggestion from looking at the tables that this might also be a problem some a few models).

While you mention using multiplicity adjustments for post-hoc tests, there are a very large number of tests performed (Table 1 includes 9 tests, Table 2 another 81 tests, Table 3 shows 9 main effects tests, Table 4 shows another 9 main effects tests, and Table 5 shows another 9 such tests). This appears to total 117, which if these were all independent, would produce around 6 false positives under a global null hypothesis. Given (from my counting) 12 statistically significant results, this suggests that it would not be at all surprising if up to half (or even more) of these tests were false positives. Given that the tests are not independent, this is a simplification, but it does illustrate that with so many tests performed, there is a high risk that several of the reported associations are spurious. Could the authors please address this matter in their statistical methods and discussion.

Given the interest in stratifying results by boys and girls (e.g. Tables 2 and 4+5), a regression-based approach would have been much more useful in allowing discussion around effect modification. Without this step being taken, the statement on Lines 211–213 is too strong in my opinion (absence of evidence is not evidence of absence, and evidence in only one strata does not indicate strata-specific differences in associations). Since no possible confounders are accounted for, which could also be done using regression-based approaches, this also lessens the interpretation of results.

If many of the children were recruited via kindergartens (as Lines 235–239 suggest and you mention again this shared factor on Lines 326–329), this would potentially introduce cluster effects that would require a different analysis approach. How were multiple children from the same families treated in the analyses? They also cannot be treated as independent in this case (sharing genetic and/or environmental factors).

While you allude to confounding as a possible explanation of the results (Lines 274–277), I think that confounding due to socio-economic status and parental lifestyle measures are overwhelming likely to explain your findings here (the first two explanations do not strike me personally as particularly plausible, although I would expect better diet quality to be associated with better anthropometry outcomes because of satiety).

I think that social desirability is a much stronger mechanism, and so threat to validity, than you acknowledge on Lines 311–312. If you assume that for the children in the study, two measures are independent and then move some points both to the right and up (assuming higher values are better), you will introduce a positive correlation simply by social desirability.

I think that the two biggest limitations of your study are the role of chance (with such a large number of statistical tests being performed and not a particularly large number of these being statistically significant) and not accounting for confounding. These warrant careful consideration around Lines 308–317.

Specific, sometimes minor, points include:

Line 19: “type of nutrition” isn’t a standard phrase and doesn’t, for me at least, capture the intention or the wording in the title. Also the order of variables differs here to that in the title and it might be worth keeping this consistent throughout the manuscript.

Line 45: “preventing” is far too strong here and in terms of energy expenditure and body composition, PA at most assists with these and I suggest appropriate qualifications be added here.

Line 47: “assures” is again far too strong.

Line 68: Reference 16, as far as I can tell, wouldn’t support this particular claim (“providing most of the necessary macronutrients in the right proportions”). Can the authors identify where in that article, this statement is made? If this reference is not able to support this statement, I recommend that the authors check all of their other references also.

Line 68: This use of “fewer” should be explicit in what it is compared to.

Line 86: The “problem” here isn’t explicitly identified.

Line 88: Do you mean “…the MOST vulnerable…” or “…A very vulnerable…”?

Lines 92–93: This a quite vague hypothesis. Did you not hypothesise the directions of associations?

Line 98: Is “the” needed here before “school enrollment”, or is a word missing at the end of the sentence?

Line 100: A comma is needed after “approval”.

Line 100: “children’s”

Line 101: Could you please give the actual participation rate and describe the reasons parents did not participate (where known)?

Lines 101–104: How many such children were excluded? What were the specific morbidities?

Lines 115–116: Could you please provide more detail here (makes and models, number of measurements and how these were combined, and any quality assurance activities).

Line 127: Without truncating the scores (not mentioned here) negative scores are possible.

Lines 127 and 128: These would need to be non-strict inequalities (otherwise 8 and 3 are not handled) as you do on Lines 141–142 (but note that it should be “8” on Line 142).

Line 138: It would be helpful to the reader to give some numerical evidence of reliability and validity here (to save them needing to check the reference).

Line 144: Note that normality of data is not needed for all of the analyses described here (normality of residuals, or data within groups, is relevant though).

Lines 146–147: There appears to be a spurious space before the comma.

Line 242: Do you mean “contradictory” here?

Line 242: It’s not entirely clear what you mean by “the country”. Do you mean in Croatia?

Line 252: Spurious space(s).

Lines 297–298: While you are generally careful around causal language, I think you go too far here in using “suggesting” when I think there are much simpler and more plausible explanations available for your results (see comments on confounding above).

Line 302: Spurious space here.

Line 304: A z-score would be much more useful here.